# Pollen Exposure and Cardiopulmonary Health Impacts in Adelaide, South Australia

**DOI:** 10.3390/ijerph19159093

**Published:** 2022-07-26

**Authors:** Monika Nitschke, David Simon, Keith Dear, Kamalesh Venugopal, Hubertus Jersmann, Katrina Lyne

**Affiliations:** 1Department for Health and Wellbeing, Adelaide, SA 5000, Australia; david.simon@sa.gov.au; 2School of Public Health, University of Adelaide, Adelaide, SA 5000, Australia; keith.dear@adelaide.edu.au; 3Wellbeing SA, Adelaide, SA 5000, Australia; kamalesh.venugopal@sa.gov.au; 4Department of Thoracic Medicine, Royal Adelaide Hospital, Adelaide, SA 5000, Australia; hubertus.jersmann@sa.gov.au; 5School of Medicine, University of Adelaide, Adelaide, SA 5000, Australia

**Keywords:** pollen count, cardiovascular health, COPD, lower respiratory, time series study

## Abstract

(1) Background: Limited research has suggested that cardiopulmonary health outcomes should be considered in relation to pollen exposure. This study sets out to test the relationship between pollen types (grasses, trees, weeds) and cardiovascular, lower respiratory and COPD health outcomes using 15 years (2003–2017) of data gathered in Adelaide, South Australia; (2) Methods: A time-series analysis by months was conducted using cardiopulmonary data from hospital admissions, emergency presentations and ambulance callouts in relation to daily pollen concentrations in children (0–17) for lower respiratory outcomes and for adults (18+). Incidence rate ratios (IRR) were calculated over lags from 0 to 7 days; (3) Results: IRR increases in cardiovascular outcomes in March, May, and October were related to grass pollen, while increases in July, November, and December were related to tree pollen. IRRs ranged from IRR 1.05 (95% confidence interval (CI) 1.00–1.10) to 1.25 (95% CI 1.12–1.40). COPD increases related to grass pollen occurred only in May. Pollen-related increases were observed for lower respiratory outcomes in adults and in children; (4) Conclusion: Notable increases in pollen-related associations with cardiopulmonary outcomes were not restricted to any one season. Prevention measures for pollen-related health effects should be widened to consider cardiopulmonary outcomes.

## 1. Introduction

There is convincing evidence that pollen exposure can affect the health of people with asthma, but other serious health outcomes due to daily variation in pollen have been less investigated [1].

As far back as the year 2000, an ecological time series study raised the possibility that pollen can contribute to cardiovascular disease (CV), chronic obstructive pulmonary disease (COPD) and pneumonia-related mortality [2]. This study, conducted in the Netherlands, has since been followed by only three other studies, all corroborating the potential effect of pollen on CV-related mortality when exploring daily levels of pollen in relation to daily health outcomes over several years. One of the studies also reported on respiratory-related mortality in relation to pollen-based on 10 years of population data from Helsinki [3,4,5]. 

In these studies, exposure to pollen resulted in a 5.5% to 40% increase in cardiovascular mortality, while respiratory mortality increased by up to 78%, depending on the type of pollen and concentrations measured. These increases in mortality rates are substantial and of especial importance considering that with ongoing climate change, accompanied by increases in carbon dioxide, warming and associated lengthening of growing seasons, pollen concentrations are expected to rise. Increasing pollen exposure has already been documented in various locations around the world [6]. Studies synthesising climate change-related threats to public health have warned about prospective increases in cardiopulmonary health impacts due to pollen, air pollution, and synergistic effects of both [7,8]. 

Biological mechanisms have been proposed in support of the cardiovascular and respiratory health effects of pollen identified through epidemiological studies, although the exact pathways still need to be explored [8]. There are plausible mechanisms that can be speculated on. In a mechanism analogous to pollen-related asthma response, it is likely that cardiopulmonary-related health effects arise due to immune defence reactions occurring in atopic individuals with endpoints inside and outside the respiratory system, including the heart and the associated circulatory system [9]. The possible nexus between allergenic activity and cardiopulmonary disease concerns due to the high prevalence of cardio-respiratory disease and atopy [10,11]. 

Considering that there are limited empirical studies in relation to pollen-related CV disease, lower respiratory disease (bronchitis), and COPD, this study aims to explore daily relationships between pollen and morbidity related to these health outcomes using hospitalisations, emergency presentations and ambulance callouts based on data gathered in Adelaide, South Australia (SA) between 2003–2017. A parallel study is currently exploring the relationship between asthma and pollen.

## 2. Materials and Methods

The relationship between daily pollen counts for grasses, weeds and trees and daily CV, lower respiratory and COPD morbidity were examined using time series analysis based on the population of the Adelaide metropolitan area. The lack of pollen data outside Adelaide precluded the inclusion of regional SA in the study. Because pollen is present year-round, we explored the relationship between pollen and cardiopulmonary outcomes by month for all 12 months. We allowed for lags of up to 7 days between pollen exposure and health outcomes.

Ethics approval (HREC/18/SAH/16) was received from the Human Research Ethics Committee (HREC), SA Department for Health and Wellbeing for the conduct of the study and for access to and use of cardiopulmonary data for hospital admissions (H), emergency presentations (ED) and ambulance callouts (A), the latter from SA Ambulance Service.

### 2.1. Cardiopulmonary Data

Morbidity data using hospital admissions and emergency department presentations were extracted using the International Classification of Disease (ICD) 10th version codes for CV (I20-I25), lower respiratory or acute bronchitis (J47 bronchiectasis, J20 acute bronchitis, J21 acute bronchiolitis, J22 unspecified acute lower respiratory infection) and COPD-related (J40, J41, J44) disease. For SA Ambulance Service codes for CV and COPD-related outcomes, the provisional medical diagnosis field in ambulance records was used. Lower respiratory disease was not part of the ambulance codes; therefore, only hospitalisations and ED outcomes were used. The health outcomes for hospitalisations were sourced from 1 January 2003–31 August 2017 (15 years) and for ED and for ambulance callouts from 1 January 2004 to 31 December 2017 (13 years). Daily CV and COPD-related health outcomes in relation to pollen exposure were analysed for adults only (18 years and over), and lower respiratory outcomes were analysed for children (0–17 years) and adults.

### 2.2. Pollen

Daily pollen grain count was obtained from the Adelaide Aerobiology Laboratory for 1 January 2003 to 31 December 2017. Pollen was categorised as grasses, trees, and weeds. Pollen was collected with a Hirst automatic volumetric spore trap and calculated as number of grains per cubic metre (m^3^) [12].

Specifically, air was drawn into the Hirst trap at 10 litres/minute, the rate of breathing. Glass slides were exposed for 24 h and examined at 9am. Pollen was caught on the Vaseline-coated glass surface. The volume of air contacting the glass was 0.077 m^3^; therefore, one pollen particle counted was equivalent to 13 particles per cubic meter which were used as the correction factor.

### 2.3. Air Pollution

Daily air pollutants used as possible confounders in the analysis were daily mean PM_10_ and PM_2.5_ in micrograms (µg) µg/m^3^ and daily maximum one-hour average concentrations of NO_2_ and O_3_ in ppb. The data were provided by the SA Environmental Protection Authority (EPA) from a collection site with high accuracy located 5 km west of the Adelaide central business district (CBD). 

### 2.4. Meteorological Data

Adelaide’s seasons are opposite of the northern hemisphere. It has mild winters (June–August) and warm, dry summers (December–February). Autumn extends from March to May and spring from September to November. Daily weather data were obtained from the Australian Bureau of Meteorology (BOM) Kent Town station (BOM code 23090), approximately 1 km from the Adelaide CBD. Variables included were maximum and minimum temperature (MaxT and MinT), rainfall in millimetres (mm) and daily average relative humidity.

### 2.5. Statistical Analysis

Time series regression analysis has been widely used to explore short-term relationships between environmental exposures and health outcomes over long periods of time. We followed the method by Bhaskaran et al. to control for seasonality and long-term trends [13] in the health outcomes data using Poisson regression in Stata 17 [14]. The ‘*splinegen*’ command was used to create cubic splines of time with 104 knots based on 15 years of data and 7 knots per calendar year, as suggested. Missing covariate data were imputed by linear regression using Stata’s ‘impute’ command. The percent of data imputed ranged from less than 0.1 percent for MaxT and MinT to 3.8% for PM_10_ and PM_2.5_. Imputed pollen data were 2.5%. Pollen counts were log-transformed using logpollen = log2(1 + pollen/10) to work with the positive skew of the pollen data. This transformation maps zero pollen onto zero and ten onto one so that the model coefficients measure the incidence rate ratio (IRR) between no pollen and 10 grains per m^3^. Additionally, an interaction term between month (January–December) and pollen concentration assessed the IRR by month. Correlations between the three pollen taxa within each month (36 correlations) were all positive and ranged from 0.03 between weeds and trees in June to 0.8 between grasses and weeds in January. Only 3 correlations exceeded 0.6. While pollen correlations were generally not large, pollen data were not considered independent, requiring inclusion of all three pollen taxa into the same regression analysis for mutual adjustment. 

Environmental confounders that are likely to be related to pollen and the health outcomes and change from day to day, such as air pollutants and weather variables, were included in the regression model. MaxT, MinT and relative humidity were included in the form of cubic splines with 2 degrees of freedom (df). Rainfall was log-transformed for days where rainfall was present, likewise with a cubic spline with 2 df, and a binary indicator was included for dry versus wet days. Additional confounders were day of the week and public holidays, which were included as categorical variables.

The shape of the exposure and response relationship was assessed by fitting 2 df spline curves relating IRRs to log pollen counts. Visual inspection of the 108 CV-related curves (3 outcomes, 1 age-group, 3 pollen types, 12 months), 144 lower respiratory curves (2 outcomes, 2 age-groups, 3 pollen types, 12 months), and 108 COPD curves (3 outcomes, 1 age-group, 3 pollen types, 12 months), together with their 95% confidence intervals (CI), showed little evidence of departure from log-linearity.

Short-term autocorrelation of the health outcomes was assessed by viewing the correlograms. Presence of correlation was adjusted by inclusion of the lagged Pearson residual (r). The Pearson residual is calculated by dividing the residuals of the health outcomes, *Y* − *fv*, where *Y* stands for health outcomes and *fv* for fitted values, by their standard deviation, which for Poisson data is estimated by the square root of the fitted values (*r* = (*Y* − *fv*)/√*fv*) [15]. The Pearson residual is thus standardised to have constant variance, and the inclusion of lagged residuals in the model corrects for autocorrelation. Hospitalisation and ED outcomes for both children and adults were adjusted by inclusion of a 3-day lagged Pearson residual to control for autocorrelation; no autocorrelation was observed for ambulance callouts.

Lags between exposure and health outcomes were used to assess the effects of pollen on cardiopulmonary outcomes on day zero and the next 7 days to allow for delayed associations. The maximum lag of 7 days was chosen by comparing models using the Akaike Information Criterion (AIC). Covariates were included in their lagged form if including lags significantly improved the model as assessed by the likelihood ratio test.

The unconstrained distributed lag models (all lag terms modelled together) provided eight coefficients per health outcome for each pollen type, by months for children and adults. The cumulative effects over lags were calculated as the sum of the coefficients and tested using the ‘*lincom*’ command in Stata [14,16]. In order to express the result as an incidence rate ratio (IRR), the sum of the coefficients was exponentiated, and the standard errors from the linear combinations of the estimators were used to calculate the lower and upper 95% CI for the risk ratio. Individual lag effects were examined graphically, but no consistent patterns were evident, so only the totals are reported. The short-term IRR related to health outcomes and pollen is commensurate to an increase in pollen of 10 grains/m^3^.

## 3. Results

Figure 1 depicts the annual cycles of averaged grass, weed and tree pollen from 2003–2017. They show low counts for grass and weed pollen in June, July, and August and the constant presence of tree pollen throughout the year. These results informed the data analysis between daily pollen and health outcomes by including an interaction by month across the entire year, rather than restricting the analysis to a pre-defined pollen season.

Descriptive statistics indicating the distribution [mean, minimum, maximum, standard deviation (SD), quartiles and 95th percentile] of the data for daily health outcomes, pollen species and environmental data are presented in Table 1. Correlations among covariates (pollen counts, air pollutants and weather variables), averaged over the 12 calendar months, ranged from −0.54 to +0.70, with the only correlation above 0.55 being that between PM_2.5_ and PM_10_. The total incidence of cardiovascular-related hospitalisations during the study period for adults was 76,872, for emergency presentations, 44,615, and for ambulance callouts, 75,874. Lower respiratory hospitalisations in children were 16,789, and for ED, 32,766. In adults, there were 12,639 lower respiratory hospital attendances and 20,398 ED presentations. COPD-related hospitalisations in adults were 45,432, for ED 33,212 and for ambulance callouts, 13,709 cases. There were 3614 days (66%) without rain and 1865 (34%) with rain.

For CV-related outcomes, increases in risk were observed for grass pollen-related hospitalisation with an IRR of 1.11 (95% Confidence Interval (95% CI) 1.00–1.24) and ambulance callouts with an IRR of 1.16 (95% CI 1.01–1.33) in March. In May, for hospitalisation with an IRR of 1.14 (95% CI 1.04–1.25), for ED, an IRR of 1.14 (95% CI 1.01–1.29). Grass-pollen-related callouts were also increased in October with an IRR of 1.14 (95% CI 1.01–1.27). Tree-pollen-related CV increases commenced in July with an IRR of 1.05 (95% CI 1.00–1.10) for ambulance callouts and an IRR of 1.09 (95% CI 1.03–1.16) for ED presentations. Further on, in November, ED presentations increased with an IRR of 1.25 (95% CI 1.12–1.40) and ED-related increases in IRR in December of 1.19 (95% CI 1.03–1.38). No CV increases were observed in relation to weed pollen.

Lower respiratory hospitalisations in adults in relation to grass pollen were only increased in November with an IRR of 1.29 (95% CI 1.04–1.59). In children, there was an increased risk of ED presentations in relation to weed pollen in February with an IRR of 1.42 (95% CI 1.07–1.90). Tree pollen was related to an increased IRR of 1.08 (95% CI 1.01–1.17) for hospitalisation in July and likewise for ED in September with an IRR of 1.12 (95% CI 1.01–1.24).

COPD in adults was only increased in relation to grass pollen in May with an IRR of 1.29 (95% CI 1.04–1.58). Other pollen taxa seemed to not have any relationships with COPD health outcomes in this study.

The graphs of estimated monthly IRRs depicted in Figure 2, Figure 3, Figure 4 and Figure 5 are accompanied by tables of IRRs, their 95% confidence intervals (95% CI) and associated *p*-values for adults and children in the Appendix A.

## 4. Discussion

This study explores pollen-related increases in the incidence of three categories of disease: CV disease, COPD, and lower respiratory disease. We discuss our findings in relation to each disease category in turn.

This study indicates pollen-related increases in CV disease during several months in adults in a time-series study exploring the effects of short-term variations in pollen concentrations. Typically, pollen-related health outcomes studies are timewise restricted to spring and summer. This makes sense in the Northern Hemisphere, where cold autumns and winters impede pollination. Adelaide is situated in a warm-to-mild temperate zone, and this restrictive approach is not ideal considering that pollen can be measured all year round. In the interest of identifying risk periods for the purpose of future prevention strategies, a monthly risk assessment between pollen and health outcomes was chosen. Results indicate that CV-related health effects in adults occur in relation to grass pollen in March, May, and October and for tree pollen in July, November, and December. 

To our knowledge, only four population-based studies have been published that are relevant for comparison in relation to CV outcomes and pollen [2,3,4,5], all of them indicating a positive relationship. Three of these studies indicate CV-related mortality related to pollen exposure [2,4,5]. One study initially hypothesised on the relationship between acute coronary syndrome-related hospitalisations and pollen, but the study instead indicated a higher risk of in-hospital mortality due to these outcomes in relation to preceding pollen concentrations [3]. The fourth study explored ED presentations for myocardial infarction in relation to pollen in Canada in a multicity case-crossover design, finding same-day pollen being related to a 5.5% higher risk for myocardial infarction over the whole study period and higher risks in May (16% higher risk) and June (10% higher risk) associated with days in the highest tertile of total pollen exposure [4]. This study resonates with our study as the risk increases are of a similar order of magnitude, where ED and hospitalisation risks increased between 5% to 25%, albeit for lower increases in pollen (per 10 grains/m^3^). Interestingly, the Canadian study also used risk ratios by month, finding a range of risk levels, probably due to different pollen types and different patterns of pollen release. 

While there are still few studies that have investigated the epidemiological relationship between pollen- and CV-related mortality and morbidity, the underlying biological mechanisms have recently been debated in light of the high prevalence of CV and allergic diseases. Research propositions a plausible allergen pathway leading to CV-related outcomes via excessive mediator-related inflammation originating from the interaction between allergens and IgE-antibodies residing in the respiratory tract on mast and basophil cells [8]. This allergen-based mechanism is supported by epidemiology using the US population-based National Health and Nutrition Examination Survey (NHANES) as an example, where the risk of CV-related disease was higher in people with allergic symptoms compared to the non-atopic group [17]. CV effects due to allergenic pathways are not restricted to pollen; they have also been observed in relation to food and medication-related allergies [9]. Other relevant mechanisms that should not be discounted relate to pollen as a component of particulate air pollution; accordingly, its intrusion into the respiratory system has systemic effects on the CV system based on pathways currently considered for particulate matter air pollution [18], but this type of speculation is clearly out of the realm of this study.

Increases in lower respiratory health outcomes, covering acute bronchitis, were observed in this study in relation to weed pollen in February and to tree pollen in July and September in children coinciding with high weed pollen counts and the beginning of a rise in tree pollen in the respective months. Increased lower respiratory disease in adults occurred in November related to tree pollen which concurred with an increase in CV-related disease, also related to tree pollen. While there is only one study that specifically assessed lower respiratory disease and pollen, there are several studies that explore total respiratory causes based on ICD coding with pollen [5,19,20]. A meta-analysis of six studies found an increased risk ratio of 1.01 (95% CI 1.00–1.015) for lower respiratory symptoms per 10 grains of pollen per m^3^ [19]. Mortality of total respiratory diseases was related to moderate (10–100 grains of pollen per m^3^) birch pollen and abundant (>100 grains of pollen per m^3^) alder pollen in a Finish study [5]. Furthermore, a positive Myrtaceae (includes eucalyptus trees) pollen relationship with total lower respiratory disease (an increase of 6.65% 95% CI: 3.27–10.14) outcomes was observed in a study in Darwin, Australia [20]. It is generally reported in the literature that the prevalence of lower respiratory illness is higher in children than in adults. The reason for this may be multifactorial: children are frequently exposed to viruses such as the respiratory syncytial virus for the first time, while adults have acquired immunity. In addition, the differences in lung anatomy may predispose. The peripheral airways of children, in particular infants, are narrower than in adults and more susceptible to further narrowing with mucosal inflammation. When infection occurs in the lower respiratory tract of a child, the effect is likely to be in the smaller airways, resulting in air trapping and atelectasis. 

A starting point for exploring a possible biological pathway explaining lower respiratory outcomes and pollen is a Finish study showing that lower and upper respiratory symptoms were highly related to atopic status in a population-based study of 1008 atopic and non-atopic adults [21]. In this study, serum IgE antibody levels were related to lower respiratory symptoms suggesting a role of pollen-related impaired immune response to infectious disease [21]. A recent study combining in vitro and in vivo experiments, as well as a human cohort, indeed shows evidence of a decreased antiviral defence system related to respiratory epithelium pollen exposure [22]. The same authors successfully showed a close link between COVID-19 infection rates and high pollen exposure across 31 countries around the globe [23]. They argue that COVID-19 incidence was increased by pollen due to the impeding effect on cell-specific immune regulators irrespective of sensitised or non-sensitised people.

In this study, COPD exacerbations in relation to pollen in adults were restricted to ambulance callouts in May, concurrently with grass-pollen-related CV emergency presentations and hospitalisations. The COPD and pollen relationship is consistent with two previous studies. One study conducted in the Netherlands observed a relationship between COPD mortality and grass pollen. Another study reported an association between total pollen and Myrtaceae tree pollen with COPD hospital admissions in Darwin. An overlap between COPD and asthma has been postulated involving a common allergenic IgE pathway [24,25]. Epidemiological observations support this overlap. A large US-based study indicated increased COPD exacerbations in the subgroup in relation to allergic phenotype indicators [26]. While we would have expected exacerbations of COPD with pollen increases in other months, the lack of this is probably due to an overall low number of COPD patients with an allergic pathway.

This study provides plausible risk information through the course of a year for CV, lower respiratory and COPD health outcomes in relation to three pollen types. As the number of comparison studies is sparse and vegetation and climate vary across the globe, it is important to replicate these relationships in different settings. The implications of such health outcomes are severe, considering commentary referring to increasing pollen load, moving and increasing pollen zones, and possible allergenicity changes in relation to climate change [6]. At the same time, there are increasing warnings about CV and respiratory health implications and climate change, raising awareness about the biological pathways by which the burden of disease could increase, but also about possible prevention strategies [7,8,9,27,28]. Based on the evidence so far, it is not clear whether the health effects discussed pertain to atopic or non-atopic populations or a mixture thereof. If they only relate to the atopic population, the population risk could be much higher. This study is of ecological design and cannot answer this question as no individual data are available. While this study is hypothesis creating, it is important to follow up these concerning results with more analytical study designs. The EPOCHAL study in Switzerland may provide a step forward as it intends to observe, among other measures, pollen-related blood pressure and heart rate variability in people with and without allergies over time in a panel study to explore pollen effects on CV disease [29]. 

To continue with the limitations of this study, we observed several significantly reduced monthly IRRs. This may have two reasons. Firstly, the pollen may be directly protective, which is unlikely. Secondly, and more plausibly, pollen counts may be a proxy for confounders that we do not know and therefore could not adjust for. In the same vein, confounding by an unknown variable could have also contributed to significantly increased IRRs during months with low pollen exposure. In relation to this point, it has to be noted that recent research has indicated that pollen counts do not necessarily represent the allergen potency allowing for deviations from the pollen count model [30]. Furthermore, pollen measurements were based at one location, which does not provide information on spatial variability. This can lead to misclassification of exposure and bias the result towards the null hypothesis, though a high correlation of grass pollen from nearby sites has been shown in a study in Melbourne [31]. Given that pollen exposure may reduce immune function in the respiratory tract, it would have been desirable to correlate the findings with influenza data. Unfortunately, influenza data were not available for much of the study period; hence, this analysis was not performed. However, if using lower respiratory outcomes as a proxy for influenza data during one of our regular Adelaide influenza peaks in April and May, our data did not show any detectable increase.

The strength of the study lies in the extensive daily time series data set of 15 years which allowed enough statistical power for the exploration of the parameters in question. Furthermore, daily data were not restricted to one season but all seasons by means of monthly risk estimates. The time series approach of Bhaskaran was used, which allows for adjustment of long-term and seasonal trends and can therefore focus on short-term exposures and health outcomes, as well as the inclusion of all possible confounders, with or without lags depending on the model assessment [13]. 

Further studies should concentrate on examining the type and extent of allergic sensitisation in Australia. It has been reported that Australia has the highest prevalence of self-reported allergic rhinitis and one of the highest prevalences of sensitisation to aeroallergens, but the selection of participants in this international study was primarily based in Melbourne, Victoria [32]. Furthermore, cohort studies should be implemented to focus on clinical and symptom outcomes in relation to pollen. The top priority and essential for any further studies in Australia is pollen monitoring which should be conducted all year round across geographical areas, and include the relevant pollen types, as has been called for previously [32,33,34]. 

Knowledge of the risk pattern is important for the appropriate timing and content of public health warnings regarding cardiopulmonary health in relation to pollen exposure. Traditionally, pollen-related warnings have targeted people with asthma and allergic rhinitis; the findings of this study suggest the need to broaden attention and include cardiopulmonary disease in prevention measures. Example public health messages include staying indoors while pollen counts are high or using N95 masks if exposure cannot be avoided. From an ecological perspective, it will be important to green urban and surrounding landscapes based on human health considerations. For this to happen successfully, knowledge of population-based pollen science should be advanced

## 5. Conclusions

This study follows the trajectory of pollen risk events throughout the year for clinically relevant cardiopulmonary health outcomes. The key months in adults for CV effects were March, July, and October–December, with risks of lower respiratory problems concomitant in November and of COPD in May. Grass and tree pollen were the main culprits. In children, where only lower respiratory health was assessed, February, July and September indicate increased risk relating to weed and tree pollen. This is one of the few studies that examines pollen-related risks for health outcomes other than asthma. The results show that the pollen provoked health risks are of clinical concern with implications at the population level. Pollen and health-related research must be embraced in Australia and public health strategies must be developed to ensure effective prevention.

## Figures and Tables

**Figure 1 ijerph-19-09093-f001:**
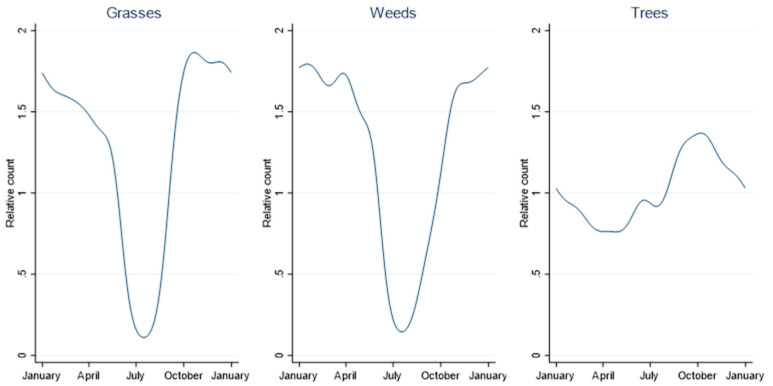
Annual cycles of pollen counts modelled using cosinor analysis with 6 harmonics.

**Figure 2 ijerph-19-09093-f002:**
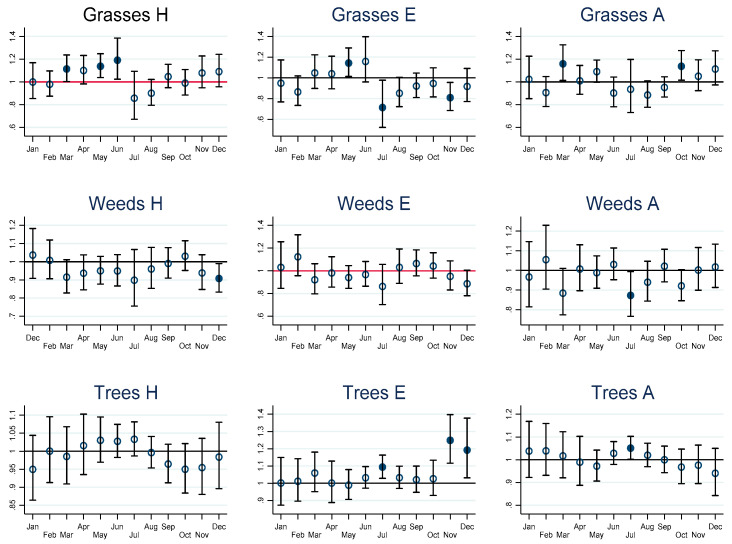
Short-term incidence rate ratios (IRRs) in adults for cardiovascular health outcomes. H = hospitalisation, E = emergency presentations, A = ambulance callouts. IRRs significantly different from 1 are highlighted using solid dots. IRRs refer to hospital admissions (H), emergency department presentations (E) and ambulance callouts (A) and pollen (Grasses, Weeds, Trees) by month. The estimated IRRs and associated 95% confidence intervals (95% CI) are based on an increase in 10 grains per cubic metre (10 grains per m^3^) compared to zero grains per m^3^. Furthermore, the IRRs are based on the sum of the coefficients of the 0–7 day lags between the exposure on day zero and the daily health outcomes from day zero to day 7.

**Figure 3 ijerph-19-09093-f003:**
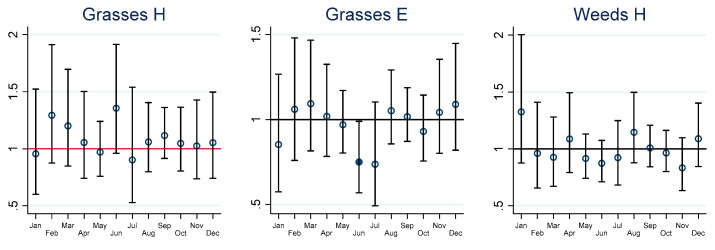
Short-term incidence rate ratios (IRRs) in adults for lower respiratory health outcomes. H = hospitalisation, E = emergency presentations. IRRs significantly different from 1 are highlighted using solid dots. IRRs refer to hospital admissions (H), emergency department presentations (E) and ambulance callouts (A) and pollen (Grasses, Weeds, Trees) by month. The estimated IRRs and associated 95% confidence intervals (95% CI) are based on an increase in 10 grains per cubic metre (10 grains per m^3^) compared to zero grains per m^3^. Furthermore, the IRRs are based on the sum of the coefficients of the 0–7 day lags between the exposure on day zero and the daily health outcomes from day zero to day 7.

**Figure 4 ijerph-19-09093-f004:**
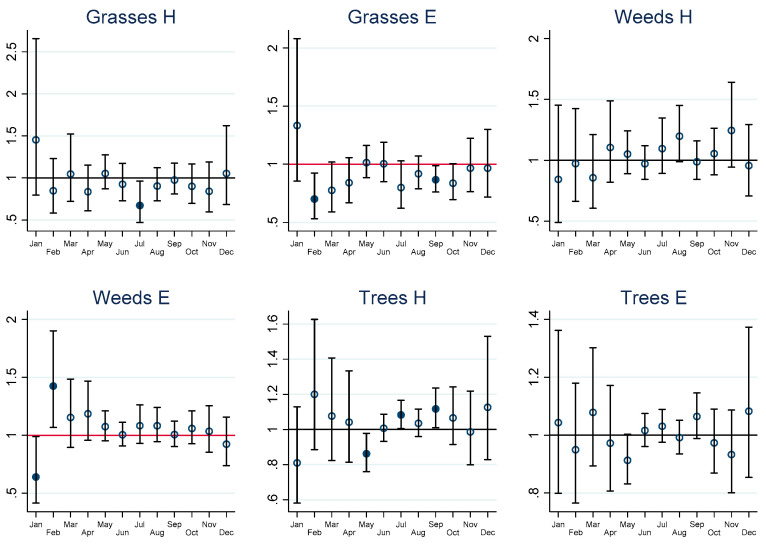
Short-term incidence rate ratios (IRRs) in children for lower respiratory health outcomes. H = hospitalisation, E = emergency presentations. IRRs significantly different from 1 are highlighted using solid dots. IRRs refer to hospital admissions (H), emergency department presentations (E) and ambulance callouts (A) and pollen (Grasses, Weeds, Trees) by month. The estimated IRRs and associated 95% confidence intervals (95% CI) are based on an increase in 10 grains per cubic metre (10 grains per m^3^) compared to zero grains per m^3^. Furthermore, the IRRs are based on the sum of the coefficients of the 0–7 day lags between the exposure on day zero and the daily health outcomes from day zero to day 7.

**Figure 5 ijerph-19-09093-f005:**
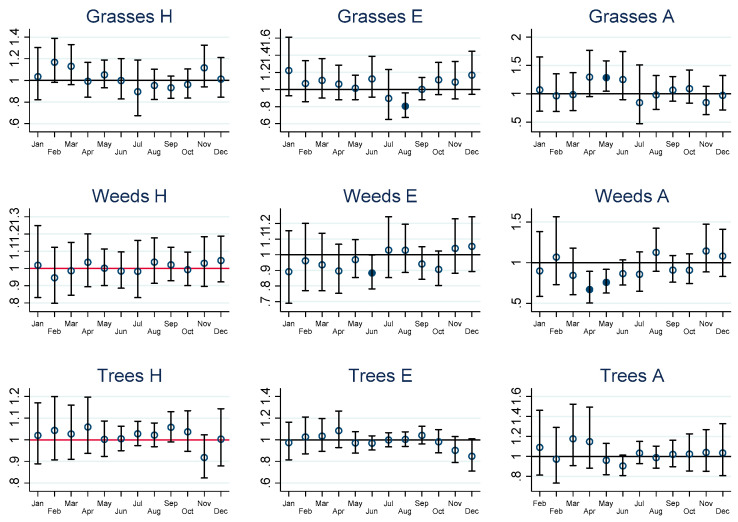
Short-term incidence rate ratios (IRRs) in adults for COPD health outcomes. H = hospitalisation, E = emergency presentations, A = ambulance callouts. IRRs significantly different from 1 are highlighted using solid dots. IRRs refer to hospital admissions (H), emergency department presentations (E) and ambulance callouts (A) and pollen (Grasses, Weeds, Trees) by month. The estimated IRRs and associated 95% confidence intervals (95% CI) are based on an increase in 10 grains per cubic metre (10 grains per m^3^) compared to zero grains per m^3^. Furthermore, the IRRs are based on the sum of the coefficients of the 0–7 day lags between the exposure on day zero and the daily health outcomes from day zero to day 7.

**Table 1 ijerph-19-09093-t001:** Descriptive data on health outcomes, pollen, and environmental confounders. (Number of days of observation for hospitalisation n = 5478; ED = 5113; ambulance n = 4935; for pollen and environmental confounder n = 5479).

Variables	Mean	Min	Max	SD	P25	P50	P75	P95
Adults								
CV-related hospitalisation	14.0	1	36	5.32	10	14	18	23
CV-related ED presentations	8.7	0	22	3.22	6	9	11	14
CV-related ambulance callouts	15.4	0	45	7.70	10	13	21	30
Lower Respiratory-related hospitalisations	2.3	0	12	1.50	1	2	3	6
Lower Respiratory-related ED presentations	4.0	0	19	2.76	2	4	6	9
COPD-related hospitalisation	8.3	0	27	3.82	6	8	11	15
COPD-related ED presentations	6.5	0	21	3.22	4	6	8	12
COPD-related ambulance callouts	2.8	0	14	2.34	1	2	4	8
Children								
Lower Respiratory-related hospitalisations	3.1	0	19	2.94	1	2	4	9
Lower Respiratory-related ED presentations	6.4	0	32	5.30	2	5	9	17
Pollen								
Grasses (grains/m^3^)	17.0	0	104	17.3	0	13	26	52
Weeds (grains/m^3^)	23.7	0	127	21.9	0	20	39	62.4
Trees (grains/m^3^)	40.6	0	273	27.0	23	36	53	88
Environmental confounders								
PM_10_ (µg/m^3^)	17.7	1	125.9	9.2	12.1	15.9	20.8	32.4
PM_25_ (µg/m^3^)	7.9	1.6	61.2	2.8	6	7.4	9.2	12.7
Max daily O_3_ (ppb)	29.0	2	105	7.8	24	28	32	44
Max daily NO_2_ (ppb)	19.0	9.5	1.0	103.0	11.0	20.0	26.0	33.0
maximum daily temperature (°C)	23.0	9.9	45.7	6.97	17.3	21.8	27.6	36.3
minimum daily temperature(°C)	12.6	0.2	33.9	5.0	9	12.1	15.5	21.8
mean daily temperature (°C)	14.5	1.3	35.5	5.6	10.6	13.8	17.5	25.4
Daily rainfall (mm)	1.5	0	75.2	4.3	0	0	0.8	8.6
Daily mean humidity (%)	69.1	5	100	20.4	58	72	86	95

CV-related IRR by month, pollen taxa and by hospitalisation, ED and ambulance callouts in adults are provided graphically in Figure 2, lower respiratory outcomes for children and adults in Figure 3 and Figure 4, and COPD outcomes for adults in Figure 5.

## Data Availability

Restrictions apply to the availability of these data. Data were obtained from several third parties including the South Australian (SA) DHW, Asthma Australia (SA branch), SA Environment Protection Authority (EPA) and the Australian Bureau of Meteorology (BoM). On reasonable request, the authors may be able to facilitate sharing the data with the permission of the third parties.

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
