# Peer review of "Pollen Exposure and Cardiopulmonary Health Impacts in Adelaide, South Australia"

_ijerph, 2022, doi:10.3390/ijerph19159093_

Round 1

Reviewer 1 Report

The manuscriptPollen exposure and cardiopulmonary health outcomes in Ade- 2 laide, South Australia” discussed the types of pollen exposure to adults and children and evaluated it in time-series analyses. I agree with the author's statement regarding limited studies available on this specific topic, and it would be valuable in current literature. However, some specific comments and suggestions should be addressed for its publication.

What are possible reasons for “high counts for grass and weed pollen in June, July, and August, and the constant presence of tree pollen throughout the year”. It can be logical somehow with weeds, but grass mostly remains like trees over the year.

How did the authors differentiate between COPD due to pollen and particulate matter exposure?

Please discuss how pollen types are age-specific? For instance, why are children more susceptible to pollen risk than adults, particularly in weed pollen and September?

Authors only provided general precautions to prevent pollen exposure, while they should provide more specific guidelines based on their findings for the study area. I may include some viable strategies to cut down/reduce pollen pollution.  

Line 73-74: How did the author assure this exposure day to define “We allowed for lags of up to 7 days between pollen exposure and health outcomes”.

Methodology

“pollen particle counted was equivalent to 13 particles per” please provide any reference,

Authors should provide the sampling site specification and also some meteorological detail and discuss it with corresponding changing pollen exposure intensity if there is any correlation between them.

Some typo

Please revise the manuscript thoroughly. Some typos need to correct. Some are as under:

Line 63: [br

Line 63: double space

Line 122: please remain consistent with the subscript of 10 and 2.5 in PM10 and PM2.5

Author Response

Thank you very much for taking the time to review our manuscript. We have tried hard to answer all questions.

Comments and Suggestions for Authors

The manuscript “Pollen exposure and cardiopulmonary health outcomes in Ade- 2 laide, South Australia” discussed the types of pollen exposure to adults and children and evaluated it in time-series analyses. I agree with the author's statement regarding limited studies available on this specific topic, and it would be valuable in current literature. However, some specific comments and suggestions should be addressed for its publication.

Reviewer 1 Point 1: What are possible reasons for “high counts for grass and weed pollen in June, July, and August, and the constant presence of tree pollen throughout the year”. It can be logical somehow with weeds, but grass mostly remains like trees over the year.

Response 1: According to our figure 1 grasses and weeds are low in June, July, and August, while for trees, there is an up-tick in their exposure in July and increasing over the following months. In our Australian winter months of June, July and August, grasses and weeds are still growing, but not flowering, hence not producing wind-dispersing male gametes (pollen). Grasses start to produce pollen in September. Recently, there has been some development in the studies of aerobiology. While hay fever, asthma and cardiopulmonary health effects can be linked to increasing pollen exposure, recent molecular experiments have indicated that pollen count alone may not be the ideal proxy for actual allergen exposure. First results from the multicentric European project, HIALINE (Health Impacts of airborne Allergen Information Network) have shown that the allergen content from the olive tree pollen (Ole e 1) and the birch tree pollen (Bet v 1) are different to the pollen counts. These early findings may explain why increases in pollen types affect health during specific time periods during pollen production and not during other times even though exposure is present.

Point 2: How did the authors differentiate between COPD due to pollen and particulate matter exposure?

Response 2: In this time series design (ecological study) we can only control for PM10 and PM2.5 which we did by including daily concentrations as confounders into the regression analysis. PM10 is particulate matter with a radius approximately < 10 micrometer (µm) and PM2.5 respectively with a diameter of <2.5 µm.

Point 3: Please discuss how pollen types are age-specific? For instance, why are children more susceptible to pollen risk than adults, particularly in weed pollen and September?

Response 3: Pollen in relation to respiratory health outcomes is usually undertaken by age groups because children have quite different airway responses to adults. The prevalence of wheeze in children in Australia is nearly 20%, whereas in adults the prevalence of asthma is about 8%.  As far as this study is concerned, we were only interested in lower respiratory (bronchitis) health outcomes in children and adults, and cardio and COPD in adults only. Allergy prevalence for specific pollen types may also be different in adults and children, but there are hardly any prevalence studies available testing this hypothesis. Studies like this would also have to be done by country/area as the flora can be quite different. Our study points to differences in adults and children in relation to the timing and the pollen type relating to bronchitis. Children have weed pollen related bronchitis in February, tree-pollen related bronchitis in July and September, while adults have tree pollen related bronchitis in November. Our study design cannot offer any explanation for the differences in susceptibility for children and adults, but this ecological study can provide clues for hypotheses testing in subsequent studies. Cohort studies and clinical studies by age groups would have to be conducted to explain differences in health effects by month. These studies would have to include allergy testing.

Point 4: Authors only provided general precautions to prevent pollen exposure, while they should provide more specific guidelines based on their findings for the study area. I may include some viable strategies to cut down/reduce pollen pollution.  

Response 4: Thank you for this point. We have alluded to some prevention strategies by referring to references 7-9, 27 and 28 in line 347. While we are restrained by the word limits, we have included a sentence in the last paragraph of the discussion (line 456-458 in the revised paper) just before the conclusion:

“From an ecological perspective it will be important to green urban and surrounding landscapes based on human health considerations. For this to happen successfully, knowledge of population-based pollen science should be advanced.”

Point 5: Line 73-74: How did the author assure this exposure day to define “We allowed for lags of up to 7 days between pollen exposure and health outcomes”.

Response 5: For clarification, the maximum lag of 7 days was chosen to minimise the AIC among the range of values tested (5-12). In the paper we wrote the following (in line 182-183 in the revised paper): “The maximum lag of 7 days was chosen by comparing models using the Akaike Information Criterion (AIC).”

Methodology

Point 6: “pollen particle counted was equivalent to 13 particles per” please provide any reference,

Response 6: This explanation is based on the very specific set up of the Hirst trap and how the slide was positioned in the trap in the location (line 115-119 in the revised paper). It was probably not relevant to refer to the procedure in details as the reference 12 is the rightful reference to this procedure (Hirst JM. An automatic volumetric spore trap. Ann Appl Biol 1952;39, 257-65).

Point 7: Authors should provide the sampling site specification and also some meteorological detail and discuss it with corresponding changing pollen exposure intensity if there is any correlation between them.

Response 7: We received the pollen data from Dr Alan Gale, a consultant physician (Allergy) who passed away in 2020. He provided pollen data to Asthma Australia over more than 20 years. The data, from his Adelaide Aerobiology Laboratory, was used to provide daily pollen levels for people with asthma in South Australia. The Hirst trap set up was on top of the roof of the Queen Elizabeth Hospital in Adelaide. As this is an international journal, we see no benefit to disclose this site. For your information, it is situated about 5 km to the West of the CBD and quite central to metropolitan Adelaide as a whole. We discussed the shortcoming of only one monitoring site in the paper.

It is worth noting that the paper describes the source of the meteorological data used, which is also from a site close to the CBD. Furthermore, we included a sentence about correlations among our co-variants into the results section (line 203-209 of the revised paper), please see below:

Section 2.4: “Daily weather data were obtained from the Australian Bureau of Meteorology (BOM) Kent Town station (BOM code 23090), approximately 1km from the Adelaide CBD.  Variables included were maximum and minimum temperature (MaxT and MinT), rainfall in millimetres (mm) and daily average relative humidity.”

“Correlations among covariates (pollen counts, air pollutants and weather variables), averaged over the 12 calendar months, ranged from -0.54 to +0.70, with the only correlation above 0.55 being that between PM2.5 and PM10.”

Point : Some typo

Point 8:Please revise the manuscript thoroughly. Some typos need to correct. Some are as under:

Line 63: [br

Line 63: double space

Line 122: please remain consistent with the subscript of 10 and 2.5 in PM10 and PM2.5

Response 8: Thank you for pointing out the typos. The bracket has been changed to a (parenthesis, the double space removed and the PM10 and PM2.5 subscripts are now used consistently. (line 145 and 146 in the revised manuscript)

Submission Date

20 June 2022

Date of this review

02 Jul 2022 06:28:14

Reviewer 2 Report

Thank you for having me the opportunity of reviewing your interesting article. This is the reviewer’s comments. As an emergency physician, it’s a considerable topic.

Here are some comments.

1. Line 63: disease[bronchitis ) à use the same mark. ( ) or [ ]

2. Describe the detailed climates of the study area. As you know, CV disease is well-known to have seasonality. But the readers of other country cannnot understand the climate or temperature related to the Months.

3. Your title is needed to be re-written. Your study results show the incidence of CV disease, not the outcomes. But you wrote the health outcome. If you want to comment the outcomes, the mortality or some adverse effects should be investigated.

Author Response

Thank you very much for taking the time to review our manuscript. We have tried hard to answer all questions.

Comments and Suggestions for Authors

Thank you for having me the opportunity of reviewing your interesting article. This is the reviewer’s comments. As an emergency physician, it’s a considerable topic.

Here are some comments.

Point 1. Line 63: disease[bronchitis ) à use the same mark. ( ) or [ ]

Response 1: Thank you for letting us know about the mistakes. The change from [ to ( has been made.

Point 2. Describe the detailed climates of the study area. As you know, CV disease is well-known to have seasonality. But the readers of other country cannnot understand the climate or temperature related to the Months.

Response 2: Thank you for pointing this out. We have inserted an explanation of Adelaide’s climate by season in relation to the northern hemisphere into the methods section under section 2.4 meteorological data (line 129-131 in the revised manuscript).

“Adelaide’s seasons are opposite to the northern hemisphere. It has mild winters (June-August), and warm, dry summers (December-February). Autumn extends from March to May and spring from September to November.”

Point 3. Your title is needed to be re-written. Your study results show the incidence of CV disease, not the outcomes. But you wrote the health outcome. If you want to comment the outcomes, the mortality or some adverse effects should be investigated.

Response 3: Thank you, this is a valid point. We have changed the title to: Pollen exposure and cardiopulmonary health impacts in Adelaide, South Australia”

Reviewer 3 Report

Summary: This is an interesting study which investigated cardiopulmonary health outcomes (CV, COPD and lower respiratory) in relation to various pollen exposures. The authors reported that grass pollen was associated with increased IRR in cardiovascular outcomes in March, May and October while tree pollen was associated with these outcomes in July, November and December. COPD increases related to grass pollen occurred only in May. Children had a higher risk of lower respiratory health outcomes in February (due to weed) and July and September (due to tree pollen), while adults had a higher risk of lower respiratory health outcomes in November (due to tree pollen).

Major comments

Comment #1: The authors only investigated cumulative 7-day lag (lag 0-lag 7), but not individual lags separately (lag 0, lag 1, lag 2 etc). I suggest doing this as some lags (e.g. lag 3) may be more important than others.

Comment #2: The authors have discussed the findings well, however, in some months, pollen exposure was associated with a decreased IRR in cardiopulmonary health outcomes. Can the authors explain this?

Comment #3: The authors have looked at the outcomes (hospitalisations, emergency presentations and ambulance callouts) separately, which I agree, but it would be good to also combine these outcomes, to see the health burden as a whole.

Comment #4: Weed is associated with an increased IRRs for adult lower respiratory health outcomes (emergency presentations) in June, but the levels were very low during this time (based on Figure 1) so how do the authors explain this? Could there be confounding that wasn’t adjusted for? Similar for grass pollen and cardiovascular health outcomes (hospitalisations) in June.

Comment #5: Can the authors explain why some associations were not observed in peak pollen periods? For example, levels of grass pollen are high in Nov-Jan but the associations with cardiovascular outcomes were not significant.

Comment #6: Can the authors also explain why the associations with lower respiratory outcomes would differ between children and adults?

Minor comments

Comment #1: It should be “need to still be explored” on P2, Line 55.

Comment #2: I suggest listing the lower respiratory health outcomes considered in the Methods e.g. acute bronchitis, pneumonia etc, so that it is clearer for readers.

Comment #3: More information is required on methods for pollen data collection. For example, what was the height of the trap (i.e. how many metres above ground?), how many traps were used (only one?), what was the distance between the trap and the Adelaide metropolitan area (within the area? How big is the metropolitan area? Is the trap a reasonable proxy for exposure)?

Comment #4: Interpretation of results (P5, Lines 188-189 – “The short-term IRR related to health outcomes..”) should be put in the statistical methods section instead.

Comment #5: Figure 2 footnote should be put under Figure 2, not Figure 5.

Comment #6: X-axis labels for some of the graphs are numerical (1-12), rather than Jan-Dec. Please change to ensure consistency.

Comment #7: Authors should also summarise the findings on COPD and lower respiratory health outcomes in the first paragraph of the discussion.

Author Response

Thank you very much for taking the time to review our manuscript. We have tried hard to answer all questions.

Summary: This is an interesting study which investigated cardiopulmonary health outcomes (CV, COPD and lower respiratory) in relation to various pollen exposures. The authors reported that grass pollen was associated with increased IRR in cardiovascular outcomes in March, May and October while tree pollen was associated with these outcomes in July, November and December. COPD increases related to grass pollen occurred only in May. Children had a higher risk of lower respiratory health outcomes in February (due to weed) and July and September (due to tree pollen), while adults had a higher risk of lower respiratory health outcomes in November (due to tree pollen).

Major comments

Point 1: The authors only investigated cumulative 7-day lag (lag 0-lag 7), but not individual lags separately (lag 0, lag 1, lag 2 etc). I suggest doing this as some lags (e.g. lag 3) may be more important than others.

Response 1: Thank you for this question. We examined individual lag effects from lag 0 to lag 7 but found no consistent pattern among the many combinations of outcome and month. For simplicity we therefore present only the totals over the 8 lags. If it is considered valuable, we could provide the individual lagged results as supplementary data.

Point 2: The authors have discussed the findings well, however, in some months, pollen exposure was associated with a decreased IRR in cardiopulmonary health outcomes. Can the authors explain this?

Response 2: This may have two reasons. Firstly, the pollen may be directly protective which is unlikely. Secondly, and more plausibly, pollen counts may be a proxy for confounders which we do not know and therefore could not adjust for. For example, human behaviour could play a role. For example, the cardiovascular-related IRR for ED for grass pollen and for weed pollen for ambulance are significantly decreased in July and this could be due to staying at home and resting in the coldest part in winter which could have a positive effect on cardiovascular health. To address this point we included a sentence where we discussed the limitations of the study (line 406-409 in the revised manuscript).

“To continue with the limitations of this study, we observed several significantly reduced monthly IRRs. This may have two reasons.  Firstly, the pollen may be directly protective which is unlikely. Secondly, and more plausibly, pollen counts may be a proxy for confounders which we do not know and therefore could not adjust for.”

Point 3: The authors have looked at the outcomes (hospitalisations, emergency presentations and ambulance callouts) separately, which I agree, but it would be good to also combine these outcomes, to see the health burden as a whole.

Response 3: Thank you for this suggestion, but we find this difficult to achieve now. An investigation following the burden of disease paradigm using a summary measure of disability-adjusted life years (DALY) is currently out of our area of expertise.

Point 4: Weed is associated with an increased IRRs for adult lower respiratory health outcomes (emergency presentations) in June, but the levels were very low during this time (based on Figure 1) so how do the authors explain this? Could there be confounding that wasn’t adjusted for? Similar for grass pollen and cardiovascular health outcomes (hospitalisations) in June.

Please see our combined response to point 4 and 5

Point 5: Can the authors explain why some associations were not observed in peak pollen periods? For example, levels of grass pollen are high in Nov-Jan but the associations with cardiovascular outcomes were not significant.

Response 4 and 5: Thank you for your comments. We have chosen to answer point 4 and 5 together. According to our figure 1, grass and weed pollen are low in June, July, and August, while for tree pollen, there is an up-tick in their exposure in July and increasing over the following months, so are the grass pollen. The question is, why is there no good coherence in relation to peak pollen periods?  While hay fever, asthma and cardiopulmonary health effects have been linked to pollen exposure in previous research, recent molecular experiments have indicated that pollen count alone may not be the ideal proxy for actual allergen exposure. First results from the multicentric European project, HIALINE (Health Impacts of airborne Allergen Information Network) have shown that the allergen content from the olive tree pollen (Ole e 1) and the birch tree pollen (Bet v 1) are substantially different to the pollen counts per grains/m3.  In addition to the difference in allergenicity compared to pollen count, there is also a difference in allergenicity in relation to time and place. These early findings may explain why increases in pollen types affect health during specific time periods. We have referred to this issue in the limitations section: “It has also been noted that pollen counts do not always represent the allergen content which would add to misclassifications [31].” While we do not want to add to the discussion overly, we have re-arranged the limitations section to deal with these issues:

“To continue with the limitations of this study, we observed several significantly reduced monthly IRRs. This may have two reasons. Firstly, the pollen may be directly protective which is unlikely. Secondly, and more plausibly, pollen counts may be a proxy for confounders which we do not know and therefore could not adjust for. In the same vein, confounding by an unknown variable could have contributed to significantly increased IRRs during some months with low pollen exposure. In relation to this point it has to be noted that recent research has indicated that pollen counts do not  necessarily represent the allergen potency allowing for deviations from the pollen count model [30]. Furthermore, pollen measurements were based at one location which does not provide information on spatial variability. This can lead to misclassification of exposure and bias the result towards the null hypothesis, though high correlation of grass pollen from nearby sites has been shown in a study in Melbourne [31]. Given that pollen exposure may reduce immune function in the respiratory tract, it would have been desirable to correlate the findings with influenza data. Unfortunately, influenza data was not available for much of the study period, hence this analysis was not performed. However, if using lower respiratory outcomes as a proxy for influenza data during one of our regular Adelaide influenza peaks in April and May, our data did not show any detectable increase.”

Point 6: Can the authors also explain why the associations with lower respiratory outcomes would differ between children and adults?

Response 6: It is generally reported in the literature that the prevalence of lower respiratory illness is higher in children than in adults. The reason for this may be multifactorial: children are frequently exposed to viruses such as RSV for the first time, while adults have acquired immunity. In addition, the differences in lung anatomy may predispose. The peripheral airways of children, in particular infants are narrower than in adults and more susceptible to further narrowing with mucosal inflammation. When infection occurs in the lower respiratory tract of a child the effect is likely to be in the smaller airways, resulting in air trapping and atelectasis.

Minor comments

Point 1: It should be “need to still be explored” on P2, Line 55.

Response 1: The authors have decided to leave “still” at its original place in the sentence.

Point 2: I suggest listing the lower respiratory health outcomes considered in the Methods e.g. acute bronchitis, pneumonia etc, so that it is clearer for readers.

Response 2:  We have included the relevant ICD 10 codes for our lower respiratory disease analysis in the methods in section 2.1 cardiopulmonary data.

Point 3: More information is required on methods for pollen data collection. For example, what was the height of the trap (i.e. how many metres above ground?), how many traps were used (only one?), what was the distance between the trap and the Adelaide metropolitan area (within the area? How big is the metropolitan area? Is the trap a reasonable proxy for exposure)?

Response 3: Thank you for asking: As you may already have gleaned from our paper in the limitations section in the discussion (now lines 406-413), we only had one pollen monitoring station for Adelaide which brings about limitations and may in fact steer the IRR results towards the 0-hypothesis. Exposure misclassifications can underestimate the results rather than overestimate. A study in Melbourne with three monitoring stations located afar from each other reported a high correlation between these sites and therefore provides confidence that the one station in Adelaide is a reasonable proxy. However, our article is pushing for more funding for increased monitoring all year round to progress pollen research.

We received the pollen data from Dr Alan Gale, a consultant physician (Allergy) who passed away in 2020. He provided pollen data to Asthma Australia over more than 20 years. The data, from his Adelaide Aerobiology Laboratory, was used to provide daily pollen levels for people with asthma in South Australia. The Hirst trap set up was on top of the roof of the Queen Elizabeth Hospital in Adelaide, but the monitoring has now ceased.

Point 4: Interpretation of results (P5, Lines 188-189 – “The short-term IRR related to health outcomes..”) should be put in the statistical methods section instead.

Response 4: We have noted your suggestion and have taken the sentence out of the results section and included it at the end of the statistical analysis section. (line 191-192 in the revised manuscript)

Point 5: Figure 2 footnote should be put under Figure 2, not Figure 5.

Response 5: We have moved the footnote relevant to Figure 2-5 below Figure 2 (line 258-263 in the revised manuscript)

Point 6: X-axis labels for some of the graphs are numerical (1-12), rather than Jan-Dec. Please change to ensure consistency.

Response 6: Thank you for pointing this out. We have changed the include x-axis names to Jan-Dec.

 Point 7: Authors should also summarise the findings on COPD and lower respiratory health outcomes in the first paragraph of the discussion.

Response 7: We added an introductory sentence at the beginning of the discussion section and then continue to discuss the three disease groups in turn. We have included the following lines at the beginning of the discussion section (Line 287-289 in the revised manuscript):

“This study explores pollen-related increases in the incidence of three categories of disease: CV disease, COPD, and lower respiratory disease. We discuss our findings in relation to each disease category in turn.

Reviewer 4 Report

Overall the manuscript was well written. However, the authors may improve the manuscript on the basis of following comments:

Abstract:

Background should be improved more.

In line 20, it was written eight-days lag. Inside the manuscript I also found 7 days lag. Please double check which was correct.

Conclusion should be improved more.

Materials and methods:

The authors may include equation of the model in statistical analysis section.

The study design would be more good if the authors analyze the result on whole population. Then in the subgroup analysis, the population would be divided into children(<17 years or less) and elderly population (>65 years). Because these age groups are more vulnerable. In addition, male and female population sub group analysis would give some important result on female population group. In the sensitivity analysis, different lag periods of the exposure and confounders may give some attractive result.  

Discussion and conclusion:

The present discussion fits the current version of analysis. If the authors will perform the additional analysis, the discussion will be changed on the basis of that.

Author Response

Thank you very much for taking the time to review our manuscript. We have tried hard to answer all questions.

Overall the manuscript was well written. However, the authors may improve the manuscript on the basis of following comments:

Abstract:

Point 1: Background should be improved more.

Point 1 has been answered below, together with point 3.

Point 2: In line 20, it was written eight-days lag. Inside the manuscript I also found 7 days lag. Please double check which was correct.

Response 2: Thank you for pointing this out. In relation to the lags, we have changed the wording in the abstract to over lags from 0 to 7 days” as the lags are indeed seven days. Day zero of exposure and outcome adds up to eight days, but day zero is not a lag day.

Point 3: Conclusion should be improved more.

Response 1 and 3: In relation to the abstract, we are confined to a 300-word limit set by the journal, hence we cannot go into details in relation to the background and conclusion in the abstract.

Materials and methods:

Point 4: The authors may include equation of the model in statistical analysis section.

Response 4: Since the basic form of the model is a conventional Poisson regression, we feel that little is gained by presenting this equation from standard theory. However, the full model would be cumbersome if written out, since it includes many covariates, most in the form of 2df spline terms and lag terms. The text describes the covariates we used, and we feel that listing them in mathematical form would add no useful information.

Point 5: The study design would be more good if the authors analyze the result on whole population. Then in the subgroup analysis, the population would be divided into children(<17 years or less) and elderly population (>65 years). Because these age groups are more vulnerable. In addition, male and female population sub group analysis would give some important result on female population group.

Response 5: At the beginning of our study, we were quite concerned about our statistical power as we thought we would have only small increases in risk and decided to use only two sub groups, children and adults. We received ethics clearance for aggregated health outcomes for children (0-17) and for adults (≥18). The aggregated data prevents us from undertaking sub group analysis.

Point 6: In the sensitivity analysis, different lag periods of the exposure and confounders may give some attractive result.  

Response 6: Thank you for your comment. Considering the already lengthy paper with multiple model results, as shown in Figures 2-5. It seems to us that duplicating these graphs for different lags, as a kind of sensitivity analysis, would make the paper too long.

Point 7: Discussion and conclusion:

The present discussion fits the current version of analysis. If the authors will perform the additional analysis, the discussion will be changed on the basis of that.

Response 7: We appreciate your comments on sensitivity and subgroups analysis. In our work in future, we will gladly utilise your suggestions.

Round 2

Reviewer 3 Report

Thank you for the responses. I just have a couple of additional minor comments:

Comment #1: Since you have investigated individual lagged effects, it is important to state this in the manuscript. I understand that the results may be overwhelming so you don't have to present them. Just a sentence to say that you have investigated this and that you found no association would be sufficient. 

Comment #2: Can you add this sentence following the limitation on unmeasured confounders "For example, the cardiovascular-related IRR for ED for grass pollen and for weed pollen for ambulance are significantly decreased in July and this could be due to staying at home and resting in the coldest part in winter which could have a positive effect on cardiovascular health". It would be clearer to the readers what you meant.

Comment #3: Response 6 should be put in the discussion. 

Author Response

Thank you for the second review.

Response point 1: We included the following sentence into the statistical analysis section (line 190-191):

Individual lag effects were examined graphically, but no consistent patterns were evident so only the totals are reported.

Response point 2: While this sentence would have fitted nicely if the weed and grass pollen would have beed high in July without having a relationship with cardiovascular disease, but in our case in Adelaide, July is not only the coldest month, but also has the lowest concentration of grass and weed pollen as you can see in figure 1. We therefore did not include your sentence in the manuscript.

Response point 3: We have included our response to your earlier review point 6 into the discussion around lower respiratory health effects (365-370 in the revised manuscript).

It is generally reported in the literature that the prevalence of lower respiratory illness is higher in children than in adults. The reason for this may be multifactorial: children are frequently exposed to viruses such as RSV for the first time, while adults have acquired immunity. In addition, the differences in lung anatomy may predispose. The peripheral airways of children, in particular infants are narrower than in adults and more susceptible to further narrowing with mucosal inflammation. When infection occurs in the lower respiratory tract of a child the effect is likely to be in the smaller airways, resulting in air trapping and atelectasis.
